# MOQAGPT: Zero-Shot Multi-modal Open-domain Question Answering with Large Language Models

**Le Zhang** [1,2], **Yihong Wu**[2], **Fengran Mo**[2], **Jian-Yun Nie**[2], **Aishwarya Agrawal**[1,2]

[1] Mila - Québec AI Institute
[2] Université de Montréal

{le.zhang,aishwarya.agrawal}@mila.quebec

## Abstract

Multi-modal open-domain question answering typically requires evidence retrieval from databases across diverse modalities, such as images, tables, passages, etc. Even Large Language Models (LLMs) like GPT-4 fall short in this task. To enable LLMs to tackle the task in a zero-shot manner, we introduce MOQAGPT[1], a straightforward and flexible framework. Using a *divide-and-conquer* strategy that bypasses intricate multi-modality ranking, our framework can accommodate new modalities and seamlessly transition to new models for the task. Built upon LLMs, MOQAGPT retrieves and extracts answers from each modality separately, then fuses this multi-modal information using LLMs to produce a final answer. Our methodology boosts performance on the MMCoQA dataset, improving F1 by +37.91 points and EM by +34.07 points over the supervised baseline. On the MultiModalQA dataset, MOQAGPT surpasses the zero-shot baseline, improving F1 by 9.5 points and EM by 10.1 points, and significantly closes the gap with supervised methods.

## 1 Introduction

Large Language Models (LLMs) including Chat-GPT (OpenAI, 2022b), LLaMA (Touvron et al., 2023), PaLM2 (Anil et al., 2023), and the recently developed GPT4 (OpenAI, 2022c), have fundamentally transformed the manner in which humans interact with machines. Due to their vast knowledge repositories and chain-of-thought reasoning capability (Wei et al., 2023), these models have proven to be capable of providing answers to a broad range of questions across domains, without the need for training on specific tasks. Nevertheless, these models face two significant challenges. First is the issue of **hallucination** (Li et al., 2023), attributable to the fact that LLMs store their knowledge in their parameters. Hallucination can seriously hamper

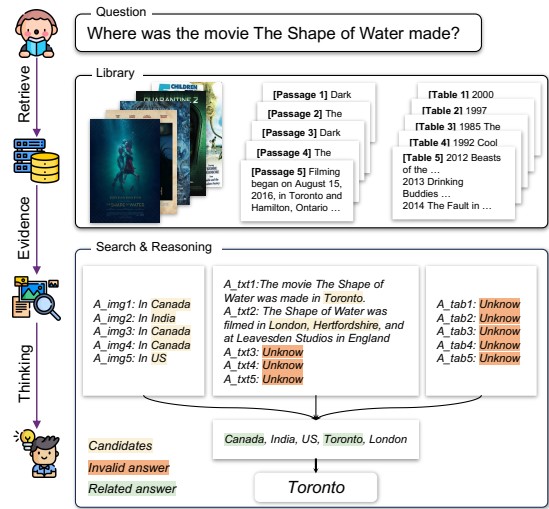

Figure 1: An illustration of how human adopt *divide-and-conquer* strategy to answer multimodal open-domain question

the accuracy and reliability of question answering, as it can introduce plausible, yet incorrect information, thus exacerbating the problem. Second, while LLMs are designed to process the text modality only, there are numerous other non-textual sources of information, such as images, videos, audios, and tables, that could provide suitable answers to most real-world questions. Some queries may even require a synthesis of information from across different modalities for accurate responses. Thus, the inability to process non-textual inputs restricts the effectiveness of current LLMs.

Consider an example outlined in fig. 1 where the question asked is, *"Where was the movie 'The Shape of Water' made?"*. To effectively answer this question, a human would employ a *divide-and-conquer* approach. This strategy involves first retrieving relevant documents such as the movie poster, news reports, and table records for *The Shape of Water*. The individual would then derive the answer from the obtained references that might include terms like *Canada*, *Toronto*, or *London, Hertfordshire in England*. Comprehensive rea-

---

[1]Our codebase is available at https://github.com/lezhang7/MOQAGPT.

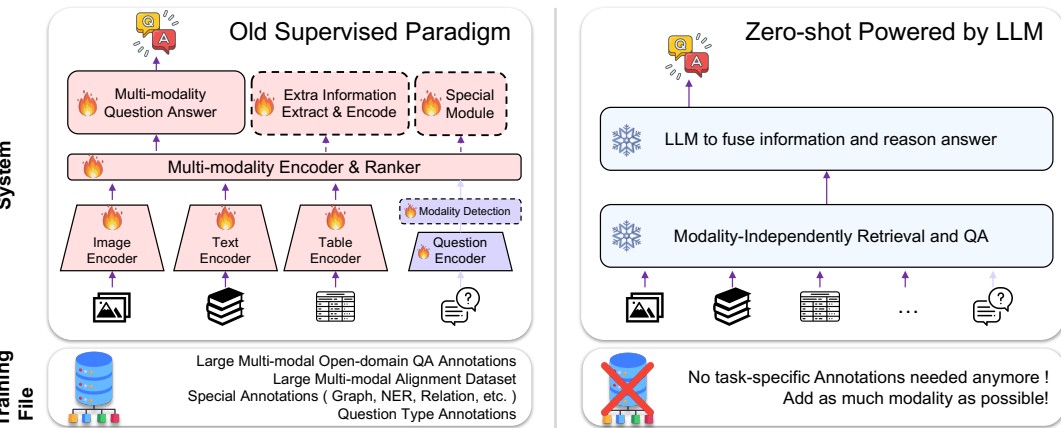

Figure 2: **Comparison of two paradigmns for multimodal open-domain question answering.** Fire symbol indicates modules require training, ice symbol indicates forzen models.

soning would be applied to all potential answers, for instance, recognizing that *Toronto* is related to *Canada*. Given this relationship, and the lack of strong ties between *London, Hertfordshire in England* and other candidate answers, *Toronto* is deemed the most probable answer and is selected as the final response.

However, many existing models rely on a *joint* strategy (Liu et al., 2023; Li et al., 2022; Yang et al., 2022; Chen et al., 2022) where they attempt to retrieve and rank all modalities by training a joint embedding space. This approach, despite its application, has several shortcomings. Firstly, the joint strategy lacks flexibility, necessitating retraining when new models and modalities are introduced. Secondly, it poses considerable difficulties to train a joint embedding space and rank references encompassing more than two modalities. Although some *divide-and-conquer* models have been proposed (Talmor et al., 2021), they also come with limitations. These models require training a question type classifier, followed by the training of various question-answering models. Each of these stages requires a significant amount of annotations, thus presenting a considerable challenge in their development and implementation.

To enable LLMs to solve this task in a zero-shot manner, we propose **M**ulti-modal **O**pen-domain **Q**uestion **A**nswering GPT (MOQAGPT). MO-QAGPT utilizes a *divide-and-conquer* approach and employs robust models to extract answers from various modalities. It further leverages LLMs as a reasoning mechanism, applying in-context learning to process the extracted information and generate the final response. Compared with the traditional supervised methods, our framework, as depicted in

fig. 2, has three main advantages: **Flexibility**: The MOQAGPT operates in zero-shot mode without relying on joint representation or inference, which allows easy replacement of individual modules with more advanced ones as they become available. Furthermore, it can accommodate a wider range of modalities and eliminates the need to curate a multimodal open-domain dataset for training. **Trustworthiness**: The framework's responses are based on the retrieved results. Consequently, each answer can be traced back to its original source, thus making the model more trustworthy and reducing the risk of hallucination. **Interpretability**: All intermediate outputs, including the retrieval results, candidate answers, and the final reasoning for answer synthesis, are produced in natural language, rendering the process of answering open-domain questions transparent and interpretable.

We corroborate these advantages by presenting experimental results on two multi-modal open-domain question answering (MMOQA) datasets: MMCoQA (Li et al., 2022) and MultModalQA (Talmor et al., 2021). Both datasets require questions to be answered based on information retrieved from text, image, and table references. We conducted experiments using several of the latest models and demonstrated that our method is effective across all of them, highlighting our framework's flexibility. To demonstrate the trustworthiness of our method, we compared our outputs to those produced by directly querying LLMs. Our outputs are less prone to hallucination, making them more trustworthy. Lastly, we examined several success and failure cases of our method. Thanks to the interpretable nature of our framework, we could identify the sources of errors. Overall, our method exhibits

robust zero-shot performance on both datasets, underscoring its promising potential.

Our contributions in this paper are threefold: (1) We propose MOQAGPT, a simple and effective framework, which is the first to enable LLMs to tackle multi-modal open-domain queries in a zero-shot setting. (2) We conduct extensive experiments involving multiple LLMs and Vision-Language Models (VLMs), thus validating the effectiveness of our approach. (3) We present empirical evidence that LLMs are capable of efficiently addressing MMOQA tasks when paired with other modalities. Furthermore, we demonstrate that replacing each module with its superior version enhances performance, establishing this as a foundational framework for future zero-shot question-answering systems.

## 2 Related Work

**Multi-modal Open-domain QA** MMOQA represents a challenging yet realistic task that is crucial for all future automated question-answering systems. This task necessitates the retrieval of pertinent references, after which answers are extracted from these references. This often involves complex processes such as modality selection and cross-modal reasoning. In light of this, several datasets have been introduced to benchmark the development of solutions in this area, such as Many-ModalQA (Hannan et al., 2020), HYBRIDQA (Chen et al., 2020), WebQA (Chang et al., 2022), MultiModalQA (Talmor et al., 2021), and MM-CoQA (Li et al., 2022).

Earlier works in this field focus on model training. These include methodologies for joint embedding of multiple modalities (e.g. MAE (Li et al., 2022) and ManyModelQA (Hannan et al., 2020)), the structured knowledge and unified retrieval-generation based method SKURG (Yang et al., 2022), and the Multimodal Graph Transformer (He and Wang, 2023), which employs a graph-based quasi-attention mechanism for integrating multi-modal graph information. To the best of our knowledge, we are the first to introduce a zero-shot method for multi-modal open-domain question answering, marking a significant contribution.

**LLM-based Modular Systems** The development of modular neural networks, rooted in biology and neuroscience, can be traced back to the 1990s (Azam, 2000; Auda and Kamel, 1999). Before the rise of LLMs, modular neural networks

like (Andreas et al., 2016, 2015) aimed to handle compositional tasks. They did this by decomposing them into sub-tasks, utilizing off-the-shelf language parsers, and then learning specialized neural modules for each. However, their applicability was limited, being constrained by parser performance and the need for hand-specified module types.

The emergence of LLMs has renewed interest in this area. LLMs address the parsing challenges without necessitating additional training. Consequently, this led to the proposition of various LLM-based systems targeting an array of compositional reasoning challenges. Examples include:

Toolformer (Schick et al., 2023), which trains language models to select tools. Visual ChatGPT (Wu et al., 2023), HuggingGPT (Shen et al., 2023), and Chameleon (Lu et al., 2023), all utilizing GPT to deduce tool sequences for response generation. ViperGPT (Surís et al., 2023) and Visprog (Gupta and Kembhavi, 2022), leveraging Codex and GPT3 (Brown et al., 2020) respectively to produce python programs for visual reasoning tasks. Yet, these methods don't address MMOQA. MMOQA inherently involves multiple steps, making it apt for a modular approach. Our framework, therefore, capitalizes on LLMs to integrate and reason about information retrieved from various modalities for MMOQA. Our approach is distinct as it requires no training, setting it apart from prior works like Schick et al. (2023). It also differentiates itself from research such as Surís et al. (2023); Gupta and Kembhavi (2022); Shen et al. (2023) with its emphasis on retrieval-based question answering. Although Chameleon (Lu et al., 2023) supports both retrieval and question answering, it doesn't address the MMOQA tasks, especially those needing cross-modal information integration and reasoning. Moreover, our method operates in a zero-shot setting, avoiding the need for intermediate programs, unlike Chameleon which requires a few-shot Python intermediate program.

## 3 MOQAGPT

MOQAGPT presents a general approach to generate answers to queries using a multi-modal knowledge base collection $\mathcal{C}$, which encompasses text $\mathcal{C}_{txt}$, tables $\mathcal{C}_{tab}$, and images $\mathcal{C}_{img}$. The *divide-and-conquer* strategy is accomplished through a two-stage process.

First, the Multi-modal Question Answer Extraction stage (§3.1) extracts answer candidates from

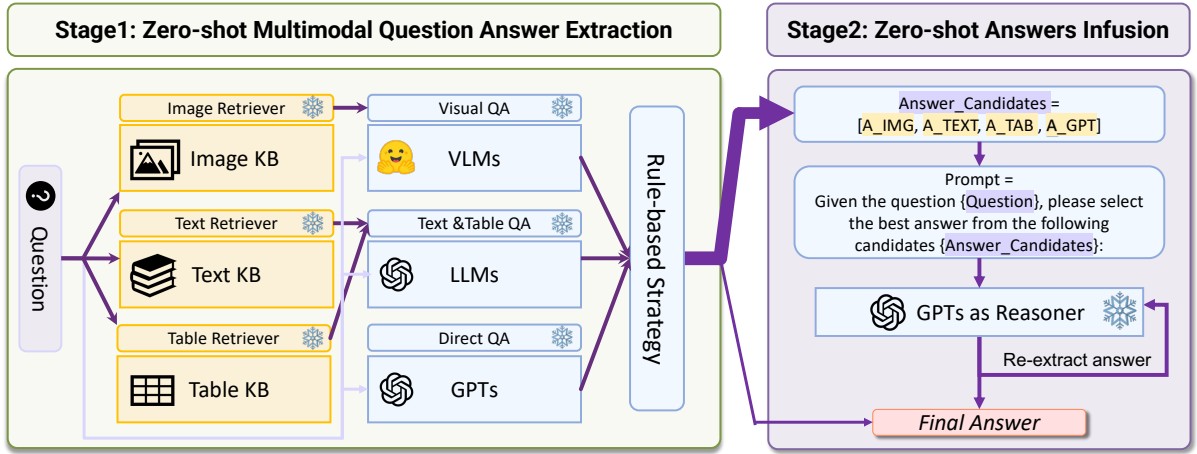

Figure 3: **Overview of MOQAGPT.** Snow symbol indicates the model is frozen.

different modalities of the knowledge base independently and utilizes rule-based strategies to sift through these responses. Second, the Answers Infusion stage (§3.2) employs LLMs' reasoning abilities to integrate information from across various modalities and select the most plausible answer. It is noteworthy that MOQAGPT operates with existing models and requires no additional training, thus exhibiting zero-shot inference capabilities. A comprehensive depiction of this methodology is provided in fig. 3, and the prompts used are described in table 1. It's worth noting that we do not perform prompt engineering due to the API cost incurred.

### 3.1 Multi-modal Question Answer Extraction

In this stage, queries are answered independently for each modality, and a strategy is employed to refine the responses. It's important to emphasize that every retrieval modality and question-answering model within our framework is frozen and can be interchanged. We delve into the details of this stage in the subsequent sections.

#### 3.1.1 Retrieval

Different modalities utilize pre-established, highly effective models for retrieval, eliminating the need to map all modalities into a joint embedding space. (i) For Text Retrieval, we use the ANCE model (Xiong et al., 2020), which adopts a dense retrieval approach to encode both passages and queries. Retrieval is then based on the cosine similarity between these encoded representations. (ii) For Image Retrieval, the renowned CLIP model (Radford et al., 2021) is employed. This zero-shot retriever, trained on web-crawled caption-image pairs using an image-text contrastive loss, has shown impressive performance in image-text retrieval benchmarks (Lin et al., 2014; Young et al., 2014). The similarity between queries and images for retrieval is determined through their inner product. (iii) For Table Retrieval, tables are typically converted into a textual format, and then language models are utilized for encoding (Herzig et al., 2020). Following this protocol, we employ OpenAI's Ada (OpenAI, 2022a), a robust embedding model, to encode linearized tables and queries, with similarity measured using the inner product. The retrieval process can be expressed as:

$$\mathcal{R}_{img} = \text{ImageRetriever}(q, \mathcal{C}_{img})$$
$$\mathcal{R}_{txt} = \text{TextRetriever}(q, \mathcal{C}_{txt})$$
$$\mathcal{R}_{tab} = \text{TableRetriever}(q, \text{linearize}(\mathcal{C}_{tab}))$$

where $\mathcal{R}$ represents the retrieved references, $q$ is the question, and $\mathcal{C}$ is the knowledge collection.

#### 3.1.2 Question Answering

Upon retrieving the references, the next step is to extract an answer from each reference based on the question $q$. (i) For Visual QA, we use vision-language models (VLMs) with robust zero-shot capabilities to generate the responses. In this step, the question is fed into these VLMs using a simple prompt: *'Question: Q Answer:'*. (ii) Since table data can be linearized into text, both Textual QA and Tabular QA can be tackled with a single LLM. As we're aiming for extractive question answering, the final response should be a concise text span from the provided input. We direct the LLMs with a specific prompt (refer to Prompt QA in table 1) for this purpose.

Moreover, we found that some questions could be addressed directly by prompting LLMs. Hence,

| Name | Prompt |
|------|--------|
| Prompt QA | *You are performing extractive question answering. Given the document: {reference} , extract a short answer to the question: {Q} from the document. If insufficient information is available to answer the question, respond with 'Unknown'. The answer should be one or two words long.* |
| Prompt Direct-QA | *Question: {questions}. Please provide a concise response, limited to one or two words, No explanation and further question. Answer:* |
| Prompt Answer-Fusion | *Given question {Q}, please select the best answer from the following candidates: {Candidates}* |
| Prompt Re-extract | *Given the question {Q}, please extract the answer span from {final answer}, without providing additional sentences or explanations. The response should be a single word.* |

Table 1: **Prompts used in MOQAGPT**

we also incorporate answers from (iii) Direct QA, where responses are obtained by directly querying the LLMs using a prompt (see Prompt Direct-QA in table 1). The overall process can be represented as:

$$\mathcal{A}_{img}^i = \text{VLM}(q, r_{img}^i)$$
$$\mathcal{A}_{txt}^i = \text{LLM}(\text{Prompt}_{QA}(q, r_{txt}^i))$$
$$\mathcal{A}_{tab}^i = \text{LLM}(\text{Prompt}_{QA}(q, \text{linearize}(r_{tab}^i)))$$
$$\mathcal{A}_{direct} = \text{LLM}(\text{Prompt}_{DirectQA}(q))$$

where $r^i$ signifies the $i^{th}$ reference from $\mathcal{R}$.

### 3.1.3 Rule-based Strategy

At this stage, we possess answer candidates derived from various modalities. Nonetheless, the autoregressive generation of responses by LLMs and VLMs can sometimes produce invalid outputs, such as *"sorry, I can't "*. Through empirical observation, we identified that: (i) The VLMs tend to consistently produce answers, even if relevant information is missing. (ii) The most accurate answer isn't necessarily found in the top-1 (most similar) retrieved reference. (iii) Using our prompts, LLMs can discern when to provide a specific answer and when to default to *"unknown"*, especially when the available information is insufficient.

With these insights, we crafted a task-agnostic, rule-based strategy to filter out invalid spans and prioritize the most likely answers generated by the LLMs and VLMs: (1) If the direct answer, $\mathcal{A}_{direct}$, is found within any of the sets $\mathcal{A}_{img}$, $\mathcal{A}_{txt}$, $\mathcal{A}_{tab}$, it's deemed reliable and is chosen as the final answer. (2) Any answer containing phrases like *"unknown"* or *"sorry"* is discarded. (3) Rather than exclusively relying on the top-1 retrieved reference, we choose the most frequent response from the top-K retrieved references. If all responses are distinct, we opt for the response from the top-1 reference.

These rules are enforced in sequence. If rule 1 isn't satisfied, we will have a curated set of valid answer candidates, denoted as $\tilde{\mathcal{A}} = \tilde{\mathcal{A}}_{img}, \tilde{\mathcal{A}}_{txt}, \tilde{\mathcal{A}}_{tab}, \tilde{\mathcal{A}}_{direct}$. This set will then be used to pinpoint the final answer span by the reasoner, detailed in the subsequent section.

### 3.2 Answer Infusion

For the majority of queries, even though it's hard to decide which modality contains the answer, the format of the answer is usually predetermined. For instance, a question like *"What color is the Santa Anita Park logo?"* should yield a color as the answer, not a date or a name. Inspired by this observation, we leverage LLMs to infer the correct answer format and select the appropriate answer from the candidates. To achieve this, we designed a prompt (refer to *Prompt Answer-Fusion* in table 1) that enables LLMs to determine the final answer. As the gold-standard answers are typically short, if the final answer contains more than three words, we guide LLMs to select the correct text span using the *Prompt Re-extract*.

## 4 Experiments & Results

This section is organized as follows. In §4.1, we outline MMOQA datasets, metrics, and baselines. We then discuss retrieval performance in §4.2, question answering in §4.3, and MMOQA results in §4.4. Lastly, we present an ablation study in §4.5 followed by a detailed case study in §4.6.

### 4.1 Implementation details

**Dataset and Metrics.** We evaluate our method on two MMOQA datasets (refer to table 2 for dataset statistics). Though they share the same references from tables, text, and images, they have different settings, questions, and answers. In all our experiments, we utilize only the **top 5 references per modality** from the knowledge collection.

| Dataset | #Questions | #Images | #Tables | #Texts |
|---------|-----------|---------|---------|--------|
| MMCoQA | 590 | 57,058 | 10,042 | 218,285 |
| MultiModalQA | 2441 | 57,058 | 10,042 | 218,285 |

Table 2: Dataset statistics

The MMCoQA dataset (Li et al., 2022) evaluates a model's proficiency in identifying the appropriate answer modality. Each question is uniquely tied to a specific modality. While the dataset employs conversational structures with historical context, we only utilized the gold question for all test set experiments to emphasize a non-conversational approach.

The MultiModalQA dataset (Talmor et al., 2021) is designed for multi-modal comprehension and multi-hop reasoning QA. In alignment with prior studies (Li et al., 2022; Talmor et al., 2021; Yang et al., 2022), we test on the development set (the test set is unlabeled and no online evaluation is available), using Exact Match and F1 as evaluation metrics. The MultiModalQA dataset already provides 1-15 reference candidates for each modality per question.

**Baselines** To the best of our knowledge, our approach is the first to enable LLMs to perform zero-shot MMOQA. For zero-shot baselines, we select Direct QA by Vicuna, OpenChat, Llama2, Chat-GPT, and GPT4[2]. Additionally, we benchmark our results against supervised methods: (i) For the MMCoQA dataset, we compare with the previous *state-of-the-art* models, MAE (Li et al., 2022), a joint embedding model trained on MMOQA datasets, and the ManyModelQA model (Hannan et al., 2020). These are the only models reported for this dataset. (ii) For the MultiModalQA dataset, we draw comparisons with the previous SOTA model SKURG (Yang et al., 2022), a structured knowledge and unified retrieval generation-based method, and the Multimodal Graph Transformer (He and Wang, 2023), a model that employs a graph-based quasi-attention mechanism to integrate multi-modal graph information.

### 4.2 Retrieval Results

For the MultiModalQA dataset, previous works directly use the gold reference set without any retrieval. We perform retrieval on the provided reference candidates (1-15 per modality) and find that Recall@5 was consistently 100%. As a result, we present the retrieval results for MMCoQA

---

[2] Details are described in Appendix appendix B.

---

| Modality | MRR | NDCG | Recall@5 | Recall@2000 |
|----------|-----|------|----------|-------------|
| *Joint (NDCG@2000)* | | | | |
| ORConvQA † | - | 2.3 | - | 19.1 |
| MAE † | - | 6.1 | - | 63.4 |
| *Divide-and-Conquer* | | | | |
| Image *(CLIP)* | 31.2 | 34.0 | 40.8 | - |
| Table *(Ada)* | 50 | 53.9 | 65.5 | - |
| Text *(ANCE)* | 35.6 | 39.0 | 49.0 | - |
| Overall | - | - | 51.0 | - |

Table 3: **Retrieval results for MMCoQA** † represents quoted results. *Joint* NDCG are computed for 2000 items. Questions are classified into categories based on the gold reference modality, scores are computed for each modality independently. *Overall* result is computed on concatenated references from all modalities, which can be viewed as Recall@5x3.

only. Each question has a designated *gold reference modality*, and we group questions based on this attribute to report a breakdown of results across modalities in table 3. This evaluation focuses solely on the retrieval of candidates.

The table indicates that our *divide-and-conquer* approach provides significant benefits in terms of effectiveness compared to the *joint* method. The prior state-of-the-art methodology, MAE, trains knowledge encoders for tables, images, and textual documents. Once trained, these knowledge encoders are frozen, and a contrastive loss function is employed to train the query encoder. This approach seek to align the embedding spaces of tables, images, and textual documents via the query embedding, without incorporating an actual multi-modality alignment. In contrast, our methodology disentangles intricate multimodal knowledge relationships by retrieving each modality independently and then assembling them using LLMs, eliminating the need for complex ranking. It registers a significant improvement, with its Recall@5x3 (51) being close to MAE's Recall@2000 (63.4).

### 4.3 Question Answering Results

We conduct question answering on the retrieved references, obtaining 5 answer candidates for each modality. The primary assessment criterion is to determine if the gold answers were present among these 5 candidates. As such, we use Recall@5 to evaluate the quality of the generated answers. Additionally, Vicuna's outputs often appear to be noisy, lengthy, and non-specific. For instance, for a question with the gold answer *"Joss Whedon"*, Vicuna might produce a response like *"1. Joss Whedon 2. David Greenwalt 3. David Boreanaz 4. Unknown"*, which complicates the extraction of the final answer, even if the recall score is high. This recall

| VLM | LLM | MMCoQA | | | | MultiModalQA | | |
| --- | --- | --- | --- | --- | --- | --- | --- | --- |
| | | Image | Table | Text | Overall | Single | Multiple | Overall |
| BLIP2 | Vicuna | 28.6 | 25.5 | 33.9 | 41.5 | 46.8 | 37.0 | 42.6 |
| InstructBLIP | Vicuna | 31.3 | 25.5 | 33.9 | 42.9 | 50.7 | 38.7 | 45.5 |
| BLIP2 | OpenChatV2 | 28.6 | 23.4 | 40.3 | 44.2 | 50.7 | 40.5 | 46.3 |
| InstructBLIP | OpenChatV2 | 31.3 | 23.4 | 40.3 | 45.4 | 53.9 | 42.0 | 48.8 |
| BLIP2 | Llam2 | 28.6 | 30.3 | 44.0 | 44.1 | 54.6 | 36.8 | 46.9 |
| InstructBLIP | Llam2 | 31.3 | 30.3 | 44.0 | 45.1 | 56.7 | 38.4 | 48.8 |
| BLIP2 | ChatGPT | 28.6 | 35.9 | 48.0 | 45.8 | 56.1 | 35.2 | 47.2 |
| InstructBLIP | ChatGPT | 31.3 | 35.9 | 48.0 | 47.6 | 58.4 | 38.1 | 49.7 |

Table 4: **Question Answering Recall@5.** We group questions based on *gold reference modality* to report results breakdown across modalities as in table 3. Similarly, we group questions answerable by a single modality or those requiring multi-modality. The *overall R@5x3* are calculated based on whether the gold answer is found within concatenated 15 answer candidate

score represents the **potential maximum** performance, as the answers from different modalities could be harmonized through LLM reasoning.

The results in table 4 indicate that VQA is the most challenging task in MMCoQA due to its low recall. In contrast, ChatGPT effectively manages textual question answering for both text and linearized tables. In the context of the MultiModalQA dataset, multi-hop reasoning or cross-modal understanding is frequently required, rendering tasks that involve multiple modalities more demanding than those relying on a single modality. Our empirical observations shows that some questions can be addressed using references from different modalities, achieving an overall Recall@5x3 of nearly 50 across both datasets.

## 4.4 Multimodal Open-domain QA Results

Following the rule-based strategy, valid results are processed with the *Prompt-Answer Infusion* and subsequently reasoned by the LLM. The results are shown in table 5 and table 6.

The *supervised* methods on the MMCoQA dataset underperform, largely due to the subpar joint retrieval results as highlighted in table 3. When MAE is provided with a gold reference, thereby eliminating the retrieval step, its performance notably improves — witnessing a 30.64 increase in F1 score and a 24.58 rise in EM. This implies that the main challenge lies in the retrieval and ranking of references. On the other hand, the *Direct QA* methods of LLMs effectively handle questions involving textual references, thanks to

| Models | | | Image | | Table | | Text | | Overall | |
| --- | --- | --- | --- | --- | --- | --- | --- | --- | --- | --- |
| VQA | Textual QA | Direct QA & Reasoner | F1 | EM | F1 | EM | F1 | EM | F1 | EM |
| | | | **Supervised** | | | | | | | |
| ORConvQA† (Qu et al., 2020) | | | - | - | - | - | - | - | 1.87 | 1.06 |
| ManyModelQA† (Talmor et al., 2021) | | | - | - | - | - | - | - | 1.82 | 0.96 |
| MAE† (Li et al., 2022) | | | - | - | - | - | - | - | 6.29 | 3.73 |
| MAE + Gold reference† | | | - | - | - | - | - | - | 36.93 | 28.31 |
| | | | **Direct QA** | | | | | | | |
| Vicuna-7B (Chiang et al., 2023) | | | 14.1 | 11.6 | 12.8 | 9.0 | 22.1 | 17.1 | 17.8 | 13.7 |
| OpenChat-v2-w-13b (Wang et al., 2023) | | | 11.3 | 10.2 | 19.7 | 15.9 | 32.5 | 25.5 | 24.1 | 19.3 |
| Llama2Chat-13b | | | 17.2 | 15.0 | 20.3 | 14.5 | 29.9 | 23.2 | 24.4 | 19.0 |
| ChatGPT (OpenAI, 2022b) | | | 17.1 | 14.3 | 25.9 | 21.4 | 45.3 | 35.6 | 33.5 | 26.8 |
| GPT4 (OpenAI, 2022c) | | | 22.0 | 16.3 | 32.1 | 26.2 | 51.2 | 44.0 | 39.2 | 32.7 |
| | | | **Zero-shot MOQAGPT** | | | | | | | |
| BLIP2 | Vicuna-7B | ChatGPT | 23.4 | 19.7 | 25.0 | 20.7 | 43.6 | 34.6 | 34.0 | 27.5 |
| BLIP2 | OpenChatV2-13b | ChatGPT | 25.9 | 22.4 | 28.0 | 23.4 | 42.8 | 35.2 | 35.0 | 29.2 |
| BLIP2 | Llama2Chat-13b | ChatGPT | 23.9 | 21.1 | 35.8 | 30.3 | 45.3 | 36.9 | 37.7 | 31.4 |
| BLIP2 | ChatGPT | ChatGPT | 24.8 | 21.1 | 38.9 | 33.1 | 47.5 | 37.6 | 39.7 | 32.4 |
| InstructBLIP | ChatGPT | ChatGPT | **28.5** | **25.9** | 36.4 | 29.7 | 46.4 | 37.6 | 39.5 | 32.7 |
| InstructBLIP | Llama2Chat-13b | Llama2Chat-13b | 22.9 | 18.4 | 31.5 | 26.4 | 41.5 | 34.5 | 34.2 | 27.8 |
| InstructBLIP | ChatGPT | Llama2Chat-13b | 23.5 | 18.9 | 32.8 | 27.2 | 42.9 | 35.4 | 35.6 | 28.9 |
| BLIP2 | ChatGPT | GPT4 | 28.3 | 22.4 | 41.3 | **35.9** | 52.8 | 45.0 | 43.9 | 37.1 |
| InstructBLIP | ChatGPT | GPT4 | 27.6 | 23.1 | **42** | 35.2 | **53.6** | **46.3** | **44.2** | **37.8** |

Table 5: **Results on MMCoQA** † represents quoted results. VQA represent models to extract answers from image, Textual QA represents models to extract answer from text and linearlized table. Direct QA & Reasoner represents model which is used to directly ask for and model to infuse information and reasoning and generate final answer

| Models | | | Single Modality | | Multi Modality | | Overall | |
|---|---|---|---|---|---|---|---|---|
| VQA | Textual QA | Direct QA & Reasoner | F1 | EM | F1 | EM | F1 | EM |
| **Supervised** | | | | | | | | |
| MGT† (He and Wang, 2023) | | | - | - | - | - | 57.7 | 52.1 |
| SKURG† (Yang et al., 2022) | | | 70.2 | 66.3 | 56.4 | 51.3 | 63.8 | 59.4 |
| **Direct QA** | | | | | | | | |
| Vicuna-7B(Chiang et al., 2023) | | | 20.3 | 17.1 | 16.4 | 11.9 | 18.6 | 14.9 |
| OpenChat-v2-w-13b (Wang et al., 2023) | | | 25.3 | 22.0 | 18.9 | 15.5 | 22.5 | 19.2 |
| Llama2Chat-13b | | | 24.6 | 21.3 | 17.3 | 13.0 | 21.5 | 17.7 |
| ChatGPT (OpenAI, 2022b) | | | 36.9 | 29.8 | 22.3 | 17.4 | 30.6 | 24.5 |
| GPT4 (OpenAI, 2022c) | | | 42.9 | 36.4 | 27.2 | 23.9 | 36.1 | 31.0 |
| **Zero-shot MOQAGPT** | | | | | | | | |
| BLIP2 | Vicuna-7B | ChatGPT | 41.3 | 34.4 | 24.7 | 20.2 | 34.2 | 28.3 |
| BLIP2 | OpenChatV2-13b | ChatGPT | 40.9 | 34.4 | 24.8 | 20.6 | 34.0 | 28.5 |
| BLIP2 | Llama2Chat-13b | ChatGPT | 43.3 | 36.8 | 27.5 | 23.2 | 36.5 | 31.0 |
| BLIP2 | ChatGPT | ChatGPT | 43.9 | 37.0 | 26.9 | 22.6 | 36.6 | 30.8 |
| InstructBLIP | ChatGPT | ChatGPT | 43.5 | 37.2 | 27.4 | 23.4 | 36.6 | 31.3 |
| InstructBLIP | Llama2Chat-13b | Llama2Chat-13b | 38.5 | 33.4 | 24.6 | 19.8 | 31.4 | 28.0 |
| InstructBLIP | ChatGPT | Llama2Chat-13b | 38.6 | 33.9 | 24.5 | 20.4 | 32.0 | 28.9 |
| BLIP2 | ChatGPT | GPT4 | 50.6 | 44.0 | 31.4 | 27.5 | 42.3 | 36.9 |
| InstructBLIP | ChatGPT | GPT4 | **54.6** | **49.1** | **33.8** | **30.5** | **45.6** | **41.1** |

Table 6: **Results on MultiModalQA** † represents quoted results.

their expansive knowledge storage. Yet, they falter for queries demanding image references, primarily due to modality limitations. Our zero-shot method outshines the *supervised* baseline because of superior retrieval, question answering, and answer infusion capabilities. It also elevates the *Direct QA* approach when grounded in retrieved results, showing up to a 6.0 F1 and 5.9 EM boost over ChatGPT and a 5.0 F1 and 5.1 EM enhancement over GPT4. Overall, our methodology exhibits a significant improvement across all tested models.

The MultiModalQA dataset provides 1-15 references for each modality, diminishing the criticality of retrieval from an extensive multi-modal knowledge base. Thus, our *divide and conquer* approach might not realize its utmost potential here. Consequently, our zero-shot method trails the supervised baselines. This is in stark contrast to MM-CoQA, where retrieval across modalities is imperative. Such foundational differences underscore the varied baseline results between the two datasets. However, given that our approach operates in a zero-shot fashion, devoid of task-specific annotations and specialized tools like question classifiers, MOQAGPT notably betters zero-shot baselines and closes the performance gap with supervised methods.

As illustrated in table 6, in comparison to *Direct QA*, our method boosts the overall metrics by 6.0 F1 and 6.8 EM over ChatGPT, and 9.5 F1

and 10.1 EM over GPT4. The most significant leap comes from the Single Modality category, underscoring the efficacy of our approach for one-hop tasks. We also register improved scores in the Multi Modality category, showcasing the ability of our GPT to amalgamate different modalities. Predictably, GPT4, employed for direct QA and reasoning, exhibits superior gains than ChatGPT across both Single/Multi Modality categories. This aligns with our hypothesis: given the task's emphasis on cross-modal reasoning, our method leans heavily on robust reasoning capabilities to merge information across modalities. Thus, the more adept the reasoner, the higher the performance. Moreover, GPTs capably filter out noise from Vicuna's output, markedly enhancing the performance against the direct QA by Vicuna for both datasets.

In conclusion, it's essential to underscore that real-world situations more closely mirror the MM-CoQA setup, where evidence isn't readily available but requires retrieval from vast repositories. In such scenarios, the strengths and merits of our method shine through, substantially surpassing supervised methods, heralding broader acceptance and use.

### 4.5 Ablation study

In our pursuit to assess the efficiency of the proposed rule-based strategy, especially its efficacy in noise mitigation, we conduct experiments on MM-CoQA. We utilize InstrucBLIP for VQA, ChatGPT

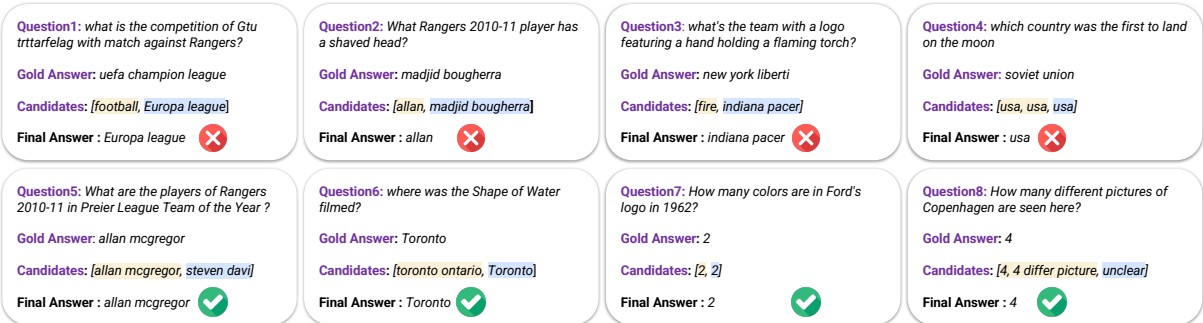

Figure 4: **Cases.** *Blue text* are Direct GPT answers. Q 1-4 are failure cases, Q 5-8 are successful cases.

| Method | Image (F1/EM) | Table (F1/EM) | Text (F1/EM) | Overall (F1/EM) | #API times |
|---|---|---|---|---|---|
| MOQAGPT | 27.1/23.1 | 42/35.2 | 53.6/46.3 | 44.2/37.8 | 423 |
| *w/o Rule1* | 26.7/22.5 | 41.6/34.9 | 52.4/44.2 | 43.3/36.2 | 590 |
| *w/o Rule2* | 24.6/19.8 | 38.6/31.4 | 49.8/43.2 | 40.8/32.9 | 423 |
| *w/o Rule3* | 26.6/21.4 | 39.4/32.7 | 50.2/43.9 | 41.8/33.4 | 423 |

Table 7: Ablations on rule-based strategy

for TextualQA, and GPT-4 for Direct QA and Reasoning. Detailed findings from these experiments are presented in table 7.

Rule 1 proves to be essential, leading to a 23% reduction in GPT activations. This obviates the need for reasoning over potentially noisy answers, thereby enhancing response accuracy and curtailing inference time. The sensitivity of LLMs to input noise, as underscored in Zhang et al. (2023), reinforces the importance of Rule 2. Excluding this rule introduces detrimental noise during the reasoning stage, adversely affecting the outcomes, as corroborated by ablation studies. Rule 3, which refines response selection by assessing the consensus among top references, is further validated through ablation study. Collectively, these findings cement the role of our rule-based strategy as a pivotal, optimized element, rather than just a rudimentary heuristic.

### 4.6 Case Study

Fig 4 presents various instances of model performance. Questions 1-4 show failures: Question 1 shows the string matching metric failing to process similar meanings. Question 2 illustrates the model's inability to choose the correct answer from the candidate list. Question 3 and 4 highlight the proposal of incorrect candidates and the existence of a knowledge bias in the model, respectively. Conversely, Questions 5-8 exemplify successes: Question 5 shows hallucination errors being rectified via grounded retrieval-based answers. Ques-

tion 6 suggests a link between retrieval-sourced answers and Direct GPT responses. Question 7 depicts that queries can be solved through retrieval methods or directly querying GPT. Lastly, Question 8 demonstrates that our framework equips LLMs with the capability to address tasks which would typically confound vanilla LLMs. More examples to demonstrate interpretability are described in appendix C.

## 5 Conclusion

In this study, we introduce the first zero-shot multi-modal open-domain question answering framework, MOQAGPT, which enables LLMs to perform the MMOQA task. This framework is flexible, accommodating new models and modalities without requiring additional training. It stands out for its trustworthiness, being grounded on reliable retrieval results, and its interpretability, which is ensured by transparent intermediate outcomes. By leveraging LLMs and VLMs, our model surpasses supervised methods in MMCoQA performance and significantly narrows the gap between zero-shot and supervised methods in multi-hop Multimodal QA datasets. Furthermore, our results indicate that models without robust knowledge storage capabilities, such as Vicuna, are less suited for this task. We hope that our approach offers some insights and servers as a general and promising framework for multi-modal open-domain question answering.

## Limitation

Central to our work is the dependency on Large Language Models (LLMs), particularly the GPT family, which being proprietary, necessitates an API call, incurring both financial and temporal costs to replicate our results. While the datasets used in our studies incurred minimal costs (2$ and 5$), larger datasets like WebQA could demand

more[3]. The consistent updates to the GPT version imply that results, while not precisely reproducible, should only improve compared to those reported in this paper. Furthermore, we provide results from open-source LLMs, ensuring reproducibility.

## Ethics Statement

The proposed method, MOQAGPT, offers substantial advances in the field of multi-modal open-domain question answering, an area of growing importance in AI. By leveraging large language models (LLMs) like GPT-4, it fosters their ability to handle tasks in a zero-shot manner and provides a robust solution for extracting and ranking answers from databases encompassing a variety of modalities such as images, tables, passages, etc.

The impact of this work extends across various sectors. By improving the efficiency and effectiveness of question-answering systems, we anticipate that this research will enhance user interactions in digital environments, streamline the retrieval of information, and significantly contribute to the development of more intuitive, accessible AI tools.

In the education sector, for example, the framework can be used to create more interactive learning systems, making it easier for students to extract accurate and comprehensive information from diverse learning materials. Additionally, in business and research domains, it could expedite data analysis by facilitating the retrieval of relevant data from vast, multi-modal databases.

While the enhancement in performance is notable, as with all AI technology, this research also presents potential societal risks. There might be an increased reliance on AI for answering questions, potentially reducing critical thinking abilities if over-relied upon. As the proposed method can cope with any modality, misuse of the technology might lead to privacy issues if it's used to retrieve sensitive information from various modalities without consent.

## Acknowledgements

We are grateful to the Mila IDT team for their technical support with the computational infrastructure. The authors acknowledge the material support of NVIDIA in the form of computational resources. During this project, Aishwarya Agrawal was supported by the Canada CIFAR AI Chair award.

---

[3]Pricing as of June 2023

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

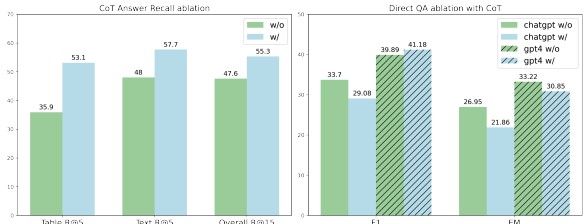

Figure 5: **QA with/without CoT.** Same metrics as table 4, VQA is InstructBLIP.

## A  Does chain-of-thought help?

CoT reasoning represents an emerging capability within LLMs. Given that we employ LLMs as our Textual QA and Reasoner, it is pertinent to examine if CoT aids in our setup. To address this, we implement a straightforward strategy, adopting a spell prompt that reads: *{reference} {question} Let's think step by step.* This results in the model generating a step-by-step reasoning response, which contemplates the reference in relation to the question, and evaluates if sufficient information is available for a response. Subsequently, we prompt GPT to extract an answer, considering the question, reasoning process, and reference with the prompt: *Reasoning:{reasoning} Question:{question} Give me a very short answer, in one or two words.* Upon conducting this process, we observe the following findings:

Firstly, CoT proves beneficial for retrieval-based question answering as demonstrated in fig. 5. The technique enables LLMs to better extract potential answers from references, significantly improving recall for table, text, and overall data. However, for Direct QA, all metrics except GPT4's F1 score decrease. This is because CoT focuses on reasoning which isn't necessary for answering general knowledge questions. We've noticed that with CoT, GPT4 tends to generate longer results, thus improving its F1 score.

| Reasoner | CoT | Recall@15 | F1 | EM |
|----------|-----|-----------|------|------|
| ChatGPT | × | 47.6 | 39.5 | 32.7 |
| ChatGPT | ✓ | 55.3 | 39.6 | 32.9 |
| GPT4 | × | 47.6 | 44.2 | 37.8 |
| GPT4 | ✓ | 55.3 | 44 | 36.8 |

Table 8: **CoT Ablation Results with Different Reasoners** VQA model is InstructBLIP and Textual QA is ChatGPT

Secondly, it's unexpected that the increased an-swer recall due to CoT does not aid in final answer extraction fig. 5. A possible reason is that CoT extracts a larger quantity of information from the reference material, both useful and irrelevant. It even attempts to provide answers when sufficient information isn't available, leading not only to include correct answers but also incorrect ones in the candidate pool. For instance, in MMCoQA, the average number of valid answer candidates with CoT is 3, compared to 2.5 without it. This addition of noise could confuse GPT4, impairing its ability to make the correct decision.

## B  Implementation details

As our method employs a zero-shot approach involving only inference, all experiments were conducted on a single A100 GPU. We employed *CLIP:ViT-B/32* for image retrieval, *text-embedding-ada-002* for table retrieval, and *ANCE-roberta* for text retrieval. We apply *BLIP2-FlanT5xl* and *InstructBLIP-vicuna-7B* for Visual Question Answering (VQA), *gpt-3.5-turbo*, *OpenChat-v2-w-13b*,*Llama2Chat-13b* and *vicuna-7B* for textual QA.

## C  Detailed Example

Our methodology is detailed in table 9 table 10 and table 11, showcasing the retrieval results, question-answering process, strategy outputs, and final answer fusion. For these examples, BLIP2 serves as the VQA model, ChatGPT as the textual QA model, and GPT4 as the reasoner and direct QA model. The retrieval results across all modalities are rational, despite several 'Unknown' instances, which are filtered out through the strategy. While the final answers are expected to be correct, they do not meet the current exact match criteria. The veracity of the results is established by grounded answer sources. For instance, in table 10, while GPT's answer is ungrounded and incorrect, our methodology provides accurate information enabling the LLM to select the correct choice.

Question: What is the competition of Gtu trttarfelag with match against Rangers?
Gold reference modality: table; Answer: uefa champion leagu

# Image Retrieval

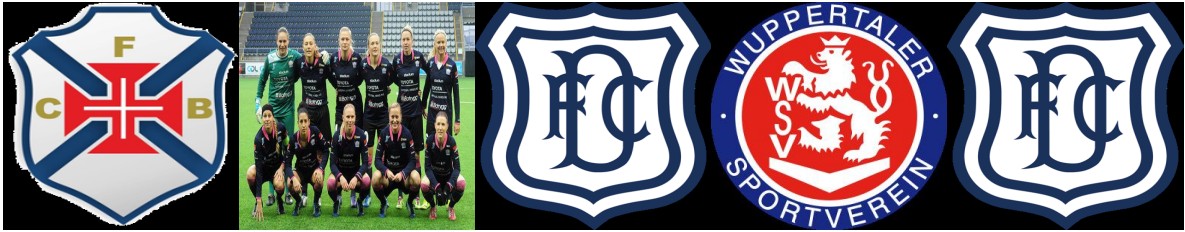

# Text Retrieval

1. Knattspyrnufélagið Ægir is an Icelandic sports club from the town of Þorlákshöfn, mainly known for its football team. The club has a football team playing in the fifth tier of Icelandic football.

2. Torfnesvöllur, known as Olísvöllurinn for sponsorship reasons, is a football stadium in Ísafjörður, Iceland and the home of Vestri and Knattspyrnufélagið Hörður. It broke ground in 1963 ...

3. Tvillingderbyt (, "The Twin Derby") is a football fixture in Stockholm, Sweden, between cross-town rivals AIK and Djurgårdens IF. Both clubs were founded in Stockholm in 1891, just three weeks ...

4. The Asturian derby (, or "Derbi astur"), is the name given to any association football match contested between Sporting de Gijón and Real Oviedo, the two biggest clubs in Asturias. The rivalry ...

5. Knattspyrnufélagið Hörður was founded on 27 May 1919 as a football club with Þórhallur Leósson being its first chairman. Its first official game was against Fótboltafélag Ísafjarðar on 17 June 1921. ...

# Table Retrieval

1. European Cup / UEFA Champions League European Cup / UEFA Champions League European Cup / UEFA Champions League European Cup / UEFA Champions ...

2. 2005–06 UEFA Cup 1Q FC Nistru Otaci 1–2 1–3 2–5 2007–08 UEFA Champions League 1Q NK Dinamo Zagreb 1–1 1–3 (aet) 2–4 2008–09 UEFA Cup ...

3. 1970–71 UEFA Cup Winners' Cup 1 Partizani 3–2 2–1 5–3 2 Real Madrid 0–2 1–0 1–2 1971–72 European Cup 1 Benfica 1–3 0–4 1–7 1972–73 ...

4. 1963–64 European Cup Preliminary round Borussia Dortmund 2–4 1–3 3–7 1964–65 European Cup Preliminary round Reipas Lahti 3–0 1–2 4–2 ...

5. 1968–69 European Cup Winners' Cup First Round FC Barcelona 0–1 0–3 0–4 1971–72 UEFA Cup First Round Legia Warszawa 1–3 0–0 1–3 1993–94 ...

# Question Answering Prompt

*You are performing extractive question answering. Given the document: {reference} , extract a short answer to the question: question from the document. If insufficient information is available to answer the question, respond with 'Unknown'. The answer should be one or two words long.*

# Image Answer

*wuppertaler, woman football, league cup, league cup, league cup*

# Text Answer

*Unknown., Unknown., Unknown., Unknown., Unknown.*

# Tabble Answer

*Unknown., Unknown., Unknown., Unknown., Unknown.*

# Chatgpt Answer

*europa leagu*

**Valid Answer Candidates(after strategy)**

*league cup,wuppertal,europa leagu*

# Answer Fusion

*Given question {Q}, please select the best answer from the following candidates: {Candidates}*

# Final Answer

*europa leagu*

Table 9: Example1, detailed results of MoqaGPT solve the task, note that there are repeated images exist in the dataset

Question: how many songs were written by john lennon and paul mccartney
Gold reference modality: text; Answer: 180

# Image Retrieval

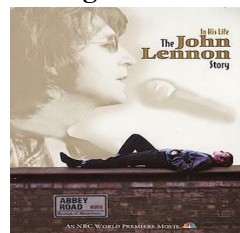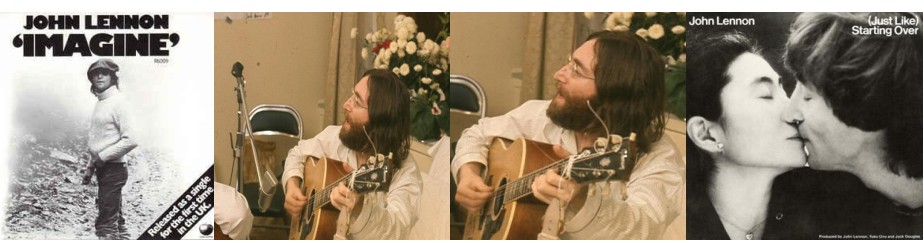

# Text Retrieval

1. *Lennon McCartney was the songwriting partnership between English musicians John Lennon (9 October 1940ŏa0 8 December 1980) and Paul McCartney ...*

2. *Lennon McCartney was the songwriting partnership between English musicians John Lennon (9 October 1940ŏa0 8 December 1980) and Paul McCartney ...*

3. *Lennon McCartney was the songwriting partnership between English musicians John Lennon and Paul McCartney of the Beatles, the partnership published approximately 180 jointly credited songs ...*

4. *Paul McCartney is an English musician who has recorded hundreds of songs over the course of his over 60-year career. As a member of the Beatles, he formed a songwriting partnership with bandmate John*

5. *Unlike many songwriting partnerships that comprise separate lyricist and composer, such as Jerry Leiber and Mike Stoller, Rodgers and Hammerstein, ...*

# Table Retrieval

1. *1 75px Dylan May 24, 1941 present Mixed-Up Confusion (1962), performed by himself 2 75px McCartney June 18, 1942 present Love Me Do/P.S. I Love You ...*

2. *1974 McGear Mike McGear 1980 The Reluctant Dog Steve Holley 1981 Somewhere in England All Those Years Ago George Harrison 1982 Tug of War ...*

3. *196? Words and Music by Paul Williams Big Seven Music Corp. 1970 Someday Man Reprise...*

4. *1 If You Be My Baby Peter Green/Clifford Adams Fleetwood Mac Mr. Wonderful (1968) 6:38 2 Long Grey Mare Peter Green Fleetwood Mac ...*

5. *Disc 1 (23971): Disc 1 (23971): Disc 1 (23971): Disc 1 (23971): Disc 1 (23971): A. I Love You Truly Carrie Jacobs Bond April 18, 1945 John Scott Trotter and His Orchestra 2:56 B. Just, ...*

# Question Answering Prompt

*You are performing extractive question answering. Given the document: {reference} , extract a short answer to the question: question from the document. If insufficient information is available to answer the question, respond with 'Unknown'. The answer should be one or two words long.*

# Image Answer

*more than one hundred, just like starting over, the john lennon story, imagine, more than one hundred*

# Text Answer

*Approximately 180., Approximately 180., Approximately 180., Unknown., Unknown.*

# Tabble Answer

*Two songs were written., Unknown., Unknown., Unknown., Unknown.*

# Chatgpt Answer

*Over 150*

Valid Answer Candidates(after strategy)

*more than 1 hundr,2 song,approxim 180 jointli credit song,over 150*

# Answer Fusion

*Given question {Q}, please select the best answer from the following candidates: {Candidates}*

# Final Answer

*approxim 180*

Table 10: Example2, detailed results of MoqaGPT solve the task

Question: What kind of coaster is shown in this photo collage of Copenhagen?
Gold reference modality: image; Answer: roller

# Image Retrieval

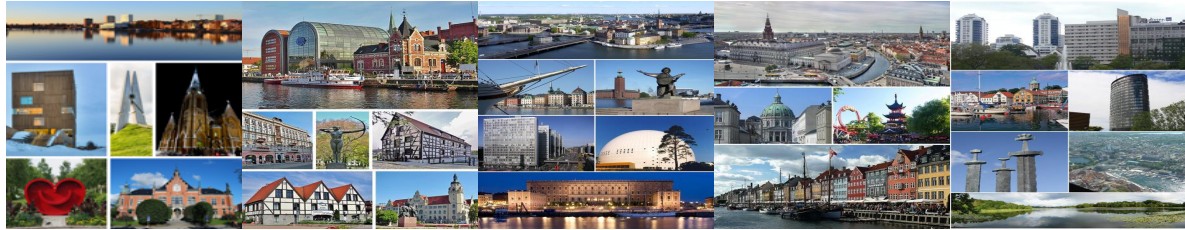

# Text Retrieval

 1. *The ride's station is located on the midway directly across from Top Thrill Dragster and was the first coaster to have inversions featuring a walkway underneath. The ride consists of ...*

2. *Kong is a steel Suspended Looping Coaster, made by Vekoma, located at Six Flags Discovery Kingdom in Vallejo, California ...*

3. *Diavlo is a steel roller coaster at Himeji Central Park in Japan which is a clone of Batman the Ride. It is one of the first ...*

4. *The first inversion in roller coaster history was part of the Centrifugal Railway of Paris, France, built in 1848. It consisted of a 43-foot (13-meter) sloping track ...*

5. *Colossus is a steel roller coaster at Thorpe Park in Surrey, England, and the park's first major attraction. It was built by Swiss manufacturers Intamin and designed ...*

# Table Retrieval

 1. *Wolverine Wildcat Michigan's Adventure United States 1988 Raging Wolf Bobs Geauga Lake United States 1988 Timber Wolf Worlds of Fun United States 1989 Hercules Dorney Park United States 1989 ...*

2. *The Bat 1987 Vekoma A Vekoma Boomerang roller coaster. It was the seventh roller coaster added to the park. The Bats train was originally from the parks Dragon Fire coaster. During the 2008 season ...*

3. *1985 Congo Carousel Robert Tidman Classic gallopers ride, operated previously at Happy Hour Amusement Park, Colwyn Bay 1986 Jungle Swings A classic chair-o-plane ride 1986 Jungle Cat ...*

4. *All Saints' Church Church of Denmark 1932 150px Drag Church Church of Denmark 1885 150px Hans Tausen's Church Church of Denmark 1924 150px Hdevang Church Church of Denmark 1885 150px ...*

5. *Ny Ellebjerg F, A, E 16 November 2006 16 November 2006 transfer to K00f8ge radial Gl. K00f8ge Landevej 2014 8 January 2005 8 January 2005 Temporary terminus; ...*

# Question Answering Prompt

*You are performing extractive question answering. Given the document: {reference} , extract a short answer to the question: question from the document. If insufficient information is available to answer the question, respond with 'Unknown'. The answer should be one or two words long.*

# Image Answer

 *roller coaster, a roller coaster, roller coaster, a wooden coaster, heart,*

# Text Answer

 *Unknown., Unknown., Unknown., Unknown., Unknown.*

# Tabble Answer

*Unknown., Unknown., Unknown., Unknown., Unknown.*

# Chatgpt Answer

*Tivoli Gardens*

# Valid Answer Candidates(after strategy)

 *roller coaster, tivoli garden*

# Answer Fusion

*Given question {Q}, please select the best answer from the following candidates: {Candidates}*

# Final Answer

*roller*

Table 11: Example3, detailed results of MoqaGPT solve the task