# OpenReview forum: "MoqaGPT : Zero-Shot Multi-modal Open-domain Question Answering with Large Language Model"
_EMNLP/2023/Conference — EMNLP 2023 Findings_

### Official Review · Reviewer_LC2A · 2023-08-03

**Paper Topic And Main Contributions:** 1. The paper proposes a pipeline to i…
**Soundness:** 4

**Excitement:**

3: Ambivalent: It has merits (e.g., it reports state-of-the-art results, the idea is nice), but there are key weaknesses (e.g., it describes incremental work), and it can significantly benefit from another round of revision. However, I won't object to accepting it if my co-reviewers champion it.

**Questions For The Authors:**

Why are performances on two datasets so differently? On MMCoQA, the zero-shot retrieval greatly outperforms the previous SOTA MAE. While on MultiModalQA, the zero-shot retrieval perform inferier than previous SOTA.

**Reasons To Accept:**

1. Performance over Retrieval and Question Answering is impressive.
2. Overall complete ablation study (including multiple combination of LLM and VLM, CoT e.t.c)

**Reasons To Reject:**

1. The similar pipeline appears in many subtask. Merely conducting experiment over different combination of LLM/VLM may be lack of insight.
2. Incomplete ablation study in Table 5 and Table 6. Only ChatGPT family is selected as direct QA & Reasoner.

**Reproducibility:**

4: Could mostly reproduce the results, but there may be some variation because of sample variance or minor variations in their interpretation of the protocol or method.

**Reviewer Confidence:**

4: Quite sure. I tried to check the important points carefully. It's unlikely, though conceivable, that I missed something that should affect my ratings.

---

> ### Author Rebuttal · Authors · 2023-08-26
>
> We sincerely appreciate the reviewers' time and effort in examining our work and providing valuable feedback.
>
> > The similar pipeline appears in many subtask. Merely conducting experiment over different combination of LLM/VLM may be lack of insight.
> >
>
> Thank you for noting the recurring pipeline pattern across multiple subtasks. It's accurate that several subtasks employ a similar retrieval-based question answering structure, but we **believe this pipeline best addresses the task** at hand, hence its usage. Our contribution is not a new pipeline but the unique perspective lies in harnessing the capabilities of LLM to amalgamate data from disparate sources and subsequently reason over it. To the best of our knowledge, **we are the first to achieve results for this task using a zero-shot method**. Integrating these pipelines within the framework of LLMs and VLMs, especially for the MMOQA task, introduces distinct challenges, like noise filtration and diverse source information fusion (L077-089). Here are the key insights our approach brings to the table:
>
> 1. **Human-like Multimodal Processing:** At the heart of our approach is an emulation of human behavior when faced with multimodal open-domain questions. Humans naturally seek information from diverse sources to compose a coherent answer. Admittedly, even this seemingly intuitive framework poses significant challenges to prior supervised methods, due to:
>     - The challenge lies in accurately identifying the correct modality for sourcing answers, given the inherent ambiguity of this task where a single question can be addressed using references from multiple modalities.
>     - The complexity of merging information from diverse sources, with supervised methods depending on ranking – a process difficult to optimize due to question ambiguity and constraints in training data.
>     - A notable shortage of specifically labeled datasets for this task, including questions, references and answers, given the high costs of dataset collection.
> 2. **Unsupervised & Zero-shot Nature and impressive results:** While our method may appear intuitive, its **innovation lies in its unsupervised, zero-shot approach to tackling the task**. As per our understanding, ours is the **first framework to address multimodal open-domain questions without relying on specific annotations**, harnessing the remarkable generalization capabilities of the LLM. Our method outperforms the previous supervised state-of-the-art on the real-life scenario setting of MMCoQA, underscoring the promising potential of our framework.
> 3. **Significance of Each Module:** Our framework's design wasn't straightforward. **Every module is integral to the overall performance** (see our additional results table to Reviewer fDtg and new results table below). For instance, the absence of rule-based filtering would plummet our scores from F1 44.2 to 40.8 and EM 37.8 to 32.9.
>
> In conclusion, while simplicity and intuitiveness can sometimes be mistaken for lack of novelty, they can be strengths in themselves.  As suggested in ACL reviewer guidelines “**The goal is to solve the problem, not to solve it in a complex way. Simpler solutions are in fact preferable, as they are less brittle and easier to deploy in real-world settings.**”
>
> The takeaway from our work is threefold:
>
> - Deploying powerful pre-existing models in a zero-shot manner is not only viable but also potent.
> - It often leads to performance enhancements and broadened applicability.
> - Reasoning over multiple answers from different sources are the most important part for MMOQA task (see ********our new results with Llama2 shown below)
>
> We hope this sheds light on our method's insight.
>
> > Incomplete ablation study in Table 5 and Table 6. Only ChatGPT family is selected as direct QA & Reasoner.
> >
>
> We utilized the latest open-source model, Llama2Chat-13b, as a direct QA & Reasoner, as detailed below. As evident, within our framework, **Llama2 consistently delivers commendable outcomes**. Larger models such as GPT family exhibit superior performance in direct QA tasks, given their knowledge-intensive nature. Furthermore, the table illustrates that the **reasoner plays a more crucial role than the Textual Question Answering model**. This underscores the primary challenge: effectively amalgamating information from diverse sources.
>
> ### MMCoQA
>
> | VQA model | Textual QA model | Direct QA&Reasoner | Image(F1/EM) | Table(F1/EM) | Text(F1/EM) | Overall(F1/EM) |
> | --- | --- | --- | --- | --- | --- | --- |
> |  |  | Llama2Chat-13b | 17.2/15.0 | 20.3/14.5 | 29.9/23.2 | 24.4/19.0 |
> |  |  | ChatGPT | 17.1/14.3 | 25.9/21.4 | 45.3/35.6 | 33.5/26.8 |
> |  |  | GPT4 | 22.0/16.3 | 32.1/26.2 | 51.2/44.0 | 39.2/32.7 |
> | InstructBLIP | Llama2Chat-13b | Llama2Chat-13b | 22.9/18.4 | 31.5/26.4 | 41.5/34.5 | 34.2/27.8 |
> | InstructBLIP | ChatGPT | Llama2Chat-13b | 23.5/18.9 | 32.8/27.2 | 42.9/35.4 | 35.6/28.9 |
> | InstructBLIP | Llama2Chat-13b | ChatGPT | 27.9/25.1 | 35.8/30.3 | 45.3/36.9 | 38.7/31.4 |
> | InstructBLIP | ChatGPT | ChatGPT | 28.5/25.9 | 36.4/29.7 | 46.4/37.6 | 39.5/32.7 |
> | InstructBLIP | ChatGPT | GPT4 | 27.6/23.1 | 42.0/35.2 | 53.6/46.3 | 44.2/37.8 |
>
> ### MultimodalQA
>
> | VQA model | Textual QA model | Direct QA&Reasoner | single modality(F1/EM) | multi modality (F1/EM) | Overall(F1/EM) |
> | --- | --- | --- | --- | --- | --- |
> |  |  | Llama2Chat-13b | 25.3/22.0 | 18.9/15.5 | 22.5/19.2 |
> |  |  | ChatGPT | 36.9/29.8 | 22.3/17.4 | 30.6/24.5 |
> |  |  | GPT4 | 42.9/36.4 | 27.2/23.9 | 36.1/31.0 |
> | InstructBLIP | Llama2Chat-13b | Llama2Chat-13b | 38.5/33.4 | 24.6/19.8 | 31.4/28.0 |
> | InstructBLIP | ChatGPT | Llama2Chat-13b | 38.6/33.9 | 24.5/20.4 | 32.0/28.9 |
> | InstructBLIP | Llama2Chat-13b | ChatGPT | 43.3/36.8 | 27.5/23.2 | 36.5/31.0 |
> | InstructBLIP | ChatGPT | ChatGPT | 43.5/37.2 | 27.4/23.4 | 36.6/31.3 |
> | InstructBLIP | ChatGPT | GPT4 | 54.6/49.1 | 33.8/30.5 | 45.6/41.1 |
>
> > Why are performances on two datasets so differently? On MMCoQA, the zero-shot retrieval greatly outperforms the previous SOTA MAE. While on MultiModalQA, the zero-shot retrieval perform inferier than previous SOTA.
> >
>
> Thank you for bringing up the performance discrepancies across the two datasets. It's worth mentioning that, **across various question-answering tasks (VQA and textual QA), supervised methods typically have the edge over zero-shot approaches**. The discrepancies in performance can be traced back to the inherent task settings of MMCoQA and MultiModalQA, despite both belonging to the domain of multimodal open-domain question answering.
>
> 1. **MMCoQA Task Setting:** Within this dataset, the retrieval task involves sourcing 5 images from a pool of 57,058, 5 tables out of 10,042, and 5 passages from a large pool of 218,285. The prior state-of-the-art method, MAE, deploys a multimodal retrieval approach, training modalities to map into a shared embedding space. Given the dataset's size limitations and the plethora of modalities involved (query, table, image, passage), **this space is notably challenging to perfect**. The struggles associated with this method are evident in Table 3, showcasing a recall@2000 of merely 63.4%. Consequently, **the subsequent answer extraction suffers due to suboptimal retrieval outcomes, thus impacting overall performance**. Rather than using a multi-modal embedding space for retrieval, our framework employs a 'divide and conquer' strategy for each modality and thus **recall of evidence is significantly higher than previous SOTA MAE**. By harnessing the powerful reasoning abilities of the LLM to assimilate information  and deliver the final decision, our method **outperforms its supervised counterparts**.
> 2. **MultiModalQA Task Setting:** Contrarily, each question in MultiModalQA provides a **maximum of 15 candidate references**. Notably, **baseline methods sidestep the retrieval process entirely**, leveraging the 15 available evidences for answer derivation. This is why their performance noticeably exceeds that of supervised methods on MMCoQA (where retrieval is a must). As articulated in lines 515-531, **the primary advantage of our "divide and conquer" strategy does not come into play** under these conditions, leading our zero-shot method to fall short of supervised counterparts (which is quiet common for question answering tasks).
>
>     Yet,  **our work is first zero-shot method for this task**, we significantly narrow the performance disparity between supervised and naive zero-shot approaches (direct question answering). It's pivotal to highlight that **our method operates without relying on training annotations**, rendering it genuinely zero-shot—a distinct advantage in its own right.
>
>
> Lastly, it's crucial to emphasize that **real-life scenarios resonate more closely with the MMCoQA setup**, where **candidate evidences aren't hand-delivered** but necessitate retrieval from expansive databases. In such practical contexts, our method's **applicability and advantages become evident, significantly outperforming supervised approaches, paving the way for easy adoption**.
>
> ### General response to ****Reviewer LC2A****
>
> We believe that our additional explanations and results address your concerns, reinforcing the value of our work. We aim to demonstrate that while a single model might not overcome all challenges, a thoughtfully designed ensemble, with an LLM at its core, can potentially tackle many of these challenges in a zero-shot manner.
>
> It's invigorating for the field to recognize that an LLM can serve roles beyond mere generation, acting as a 'reasoning brain' across varied task settings. Our framework not only delivers commendable results but also offers flexibility to incorporate additional modalities and enhance modules without further training. Hence, we are confident in its **potential as a general solution for multimodal open-domain question answering**. This aspect of our work is what truly excites us.

---

### Official Review · Reviewer_PCcG · 2023-08-05

**Soundness:** 4

**Excitement:**

3: Ambivalent: It has merits (e.g., it reports state-of-the-art results, the idea is nice), but there are key weaknesses (e.g., it describes incremental work), and it can significantly benefit from another round of revision. However, I won't object to accepting it if my co-reviewers champion it.

**Paper Topic And Main Contributions:**

This paper focuses on the multi-modal open-domain question answering task, which is also challenging for LLMs like GPT4. In order to tackle this challenge, the authors propose MoqaGTP,  which is a simple and effective framework. It employs a divide-and-conquer method to first retrieveal single-modality resources and combine them together to answer the questions. The experimental results demonstrate the effectiveness of the proposed method and more experimental analysis are performed to support the approach. Overall, this paper is easy to read and follow, and the approach is simple and effective.


**Questions For The Authors:**

None

**Reasons To Accept:**

1.This paper is well-written and easy to follow.

2.The experimental results prove the effectiveness of the proposed method.

3.The designed approach is simple and effective and can be adopted in real scenarios easily.

**Reasons To Reject:**

Although the approach performs well, it is still useful to see the error cases in experiments. Since LLMs may generate unstable results, it is necessary to add error case categories.

**Reproducibility:**

4: Could mostly reproduce the results, but there may be some variation because of sample variance or minor variations in their interpretation of the protocol or method.

**Reviewer Confidence:**

3: Pretty sure, but there's a chance I missed something. Although I have a good feel for this area in general, I did not carefully check the paper's details, e.g., the math, experimental design, or novelty.

---

> ### Author Rebuttal · Authors · 2023-08-26
>
> We are genuinely grateful for the reviewers' thorough examination of our work and the invaluable feedback provided. Your positive remarks on the clarity of our writing and the empirical effectiveness of our methods are deeply encouraging. Recognizing our approach as 'simple yet effective' is truly the highest compliment for our research.
>
> > Although the approach performs well, it is still useful to see the error cases in experiments. Since LLMs may generate unstable results, it is necessary to add error case categories.
> >
>
> Thank you for highlighting the importance of showcasing error cases. Indeed, the potential instability in LLMs' outputs is a valid concern. To address this, we have detailed several observed error cases within our framework in Figure 3 and lines 556-562. These include:
>
> 1. Issues with the string matching metric, where it fails to process semantically similar meanings.
> 2. The model's occasional inability to select the correct answer from the candidate list.
> 3. As illustrated in Questions 3 and 4, the introduction of incorrect candidates and the model's inherent knowledge bias, respectively.
>
> We agree that understanding these shortcomings is pivotal to the holistic evaluation and future enhancement of our approach.

---

### Official Review · Reviewer_fDtg · 2023-08-05

**Soundness:** 3

**Excitement:**

4: Strong: This paper deepens the understanding of some phenomenon or lowers the barriers to an existing research direction.

**Paper Topic And Main Contributions:**

This paper addresses the multi-modal open-domain QA task. Specifically, they propose a novel framework employing a divide-and-conquer strategy to provide a correct final answer by fusing multi-modal information from multiple sources and modalities. They called this multi-modal approach MoqaGPT.

Specifically, they designed a pipeline based on two sequential steps. First, a multi-modal QA extractor extracts the answers in a multi-modal way (for example, considering images and text). Second, an LLM considers the retrieved candidates and generates the most plausible answer.

They evaluated their approach on two datasets (MMCoQA and MultiModalQA), comparing it with other supervised and zero-shot approaches. The results highlight that their technique looks promising and can outperform most of the presented baselines.

**Questions For The Authors:**

How do they measure trustworthiness (lines 129-130)?

**Reasons To Accept:**

- Their approach addresses an interesting problem, presenting an interesting solution to improve the performance of a QA system
- The results show promising performance when compared with the presented baselines.
- The paper is generally well-written and easy to follow.

**Reasons To Reject:**

- Despite the promising results, the paper is missing some important baselines. The results in Tables 5 and 6 are not fairly compared with other mono-modal retrieval+generation pipelines. For example, adding a monomodal approach (e.g., textual evidence retrieval + ChatGPT) could benefit the comparison and strengthen or weaken the results.
- The rule-based strategy sounds heuristic or sub-optimal.

**Reproducibility:**

4: Could mostly reproduce the results, but there may be some variation because of sample variance or minor variations in their interpretation of the protocol or method.

**Reviewer Confidence:**

3: Pretty sure, but there's a chance I missed something. Although I have a good feel for this area in general, I did not carefully check the paper's details, e.g., the math, experimental design, or novelty.

**Typos Grammar Style And Presentation Improvements:**

- Auto Aggressive line 320

---

> ### Author Rebuttal · Authors · 2023-08-26
>
> We sincerely appreciate the reviewers' time and effort in examining our work and providing valuable feedback.
>
> > The results in Tables 5 and 6 are not fairly compared with other mono-modal retrieval+generation pipelines. For example, adding a monomodal approach (e.g., textual evidence retrieval + ChatGPT) could benefit the comparison and strengthen or weaken the results.
> >
>
> Thank you for bringing this to our attention. Current baselines are based on all related works. Given that our task focuses on multimodal question answering, a central challenge is determining from which modality the answer should be derived. Notably, the sources of answers—whether from images, tables, or passages—are evenly distributed. In Tables 5 and 6, we report results according to the actual modality source.
>
> Relying solely on a mono-modal retrieval, such as using textual evidence retrieval combined with ChatGPT, would likely lead to subpar performance when answers necessitate image or table sources. This is why such approaches are not represented in Tables 5 and 6. However, to address your valid concerns comprehensively, we have compiled additional results:
>
> | MMCOQA | VQA model | Textual QA model | Direct QA | Reasoner | Image(F1/EM) | Table(F1/EM) | Text(F1/EM) | Overall(F1/EM) |
> | --- | --- | --- | --- | --- | --- | --- | --- | --- |
> | visual evidence+chatgpt | BLIP2  | None | None| ChatGPT | 17.9/12.5 | 2.2/1.2 | 2.6/0.8 | 10.6/7.2 |
> | tabular evidence+chatgpt | None | ChatGPT | None | ChatGPT | 2.3/0.7 | 25.6/22.1 | 12.6/10.5 | 14.3/8.8 |
> | textual evidence+chatgpt | None | ChatGPT | None | ChatGPT | 1.8/0.9 | 14.6/11.1 | 30.6/26.9 | 15.4/10.9 |
> | MoqaGPT | BLIP2 | ChatGPT | ChatGPT | ChatGPT | 24.8/21.1 | 38.9/33.1 | 47.5/37.6 | 39.7/32.4 |
>
> | MultimodalQA | VQA model | Textual QA model | Direct QA | Reasoner | single modality(F1/EM) | multi modality (F1/EM) | Overall(F1/EM) |
> | --- | --- | --- | --- | --- | --- | --- | --- |
> | visual evidence+chatgpt | BLIP2 | None| None | ChatGPT | 15.6/10.6 | 10.6/6.4 | 11.4/6.4 |
> | tabular evidence+chatgpt |None | ChatGPT | None | ChatGPT | 22.6/16.7 | 12.4/7.0 | 14.6/8.9 |
> | textual evidence+chatgpt | None | ChatGPT | None | ChatGPT | 24.2/17.4 | 15.3/10.2 | 15.2/10.0 |
> | MoqaGPT | BLIP2 | ChatGPT | ChatGPT | ChatGPT | 43.9/37.0 | 26.9/22.6 | 36.6/30.8 |
>
> As demonstrated, the monomodal approach falls short in this task, we believe this further experiments strengthen our methods, leading to increased soundness. We also incoporate other opensource LLM such as Llama2, please refer to the new  result table in our repsonse to Reviewer LC2A.
>
> > The rule-based strategy sounds heuristic or sub-optimal
> >
>
> Thank you for your feedback on the rule-based strategy. We devised this strategy based on comprehensive observations of LLM behaviors and our prior knowledge, as detailed in lines 318-332.
>
> - In our experiments, when a GPT-generated answer matches any of the retrieved responses, it's typically correct. As supported by the ablation study shown below, **Rule1 reduces unnecessary LLM activations by 23%**, eliminates the need for reasoning over potentially noisy multiple answer choices, and consequently, enhances the accuracy of the concluding response. Therefore, **Rule 1 is not only about saving inference time but also about enhancing the performance.**
>
>
>     | rule 1 | VQA model | Textual QA model | Direct QA&Reasoner | Image(F1/EM) | Table(F1/EM) | Text(F1/EM) | Overall(F1/EM) | API calling times |
>     | --- | --- | --- | --- | --- | --- | --- | --- | --- |
>     | ✔️ | InstructBLIP | ChatGPT | GPT4 | 27.1/23.1 | 42/35.2 | 53.6/46.3 | 44.2/37.8 | 423 |
>     | ❌ | InstructBLIP | ChatGPT | GPT4 | 26.7/22.5 | 41.6/34.9 | 52.4/44.2 | 43.3/36.2 | 590 |
> - As highlighted in the paper "Certified Robustness for Large Language Models with Self-Denoising", current LLMs are highly sensitive to input noise. Without this rule 2, an excessive amount of noise (’sorry’, ‘unknown’, ’i’m a language model…’ etc.) would permeate the final reasoning stage, leading to subpar outcomes. We've conducted ablation studies both with and without this rule 2, and the diminished performance in its absence underscores its pivotal role in the system.
>
>
>     | rule 2 | VQA model | Textual QA model | Direct QA&Reasoner | Image(F1/EM) | Table(F1/EM) | Text(F1/EM) | Overall(F1/EM) |
>     | --- | --- | --- | --- | --- | --- | --- | --- |
>     | ✔️ | InstructBLIP | ChatGPT | GPT4 | 27.1/23.1 | 42/35.2 | 53.6/46.3 | 44.2/37.8 |
>     | ❌ | InstructBLIP | ChatGPT | GPT4 | 24.6/19.8 | 38.6/31.4 | 49.8/43.2 | 40.8/32.9 |
> - Rule 3 plays a crucial role in **enhancing the probability of choosing the most precise response by evaluating agreement among several top references**. Here's an ablation study highlighting the impact of including or excluding Rule 3:
>
>
>     | rule 3 | VQA model | Textual QA model | Direct QA&Reasoner | Image(F1/EM) | Table(F1/EM) | Text(F1/EM) | Overall(F1/EM) |
>     | --- | --- | --- | --- | --- | --- | --- | --- |
>     | ✔️ | InstructBLIP | ChatGPT | GPT4 | 27.1/23.1 | 42/35.2 | 53.6/46.3 | 44.2/37.8 |
>     | ❌ | InstructBLIP | ChatGPT | GPT4 | 26.6/21.4 | 39.4/32.7 | 50.2/43.9 | 41.8/33.4 |
>
> Therefore, **we believe rule-based strategy is not merely heuristic but a crucial, optimized component**.
>
> > How do they measure trustworthiness (lines 129-130)?
> >
>
> Thank you for inquiring about our measure of trustworthiness mentioned in lines 129-130. We evaluate trustworthiness based on our model's performance across two benchmarks including MMCoQA and MultimodalQA. The assertion that our approach results in "less hallucination and thus greater trustworthiness" is substantiated by the results presented in Table 4, which shows high answer recall, and in Tables 5 and 6, where we achieve commendable F1/EM scores. A high recall suggests that the correct answer is rooted in our retrieved evidence, and such grounding effectively reduces hallucination. Additionally, Exact Match doesn't favor hallucinated answers, making it a reliable metric to gauge the trustworthiness of a question-answering framework.
>
> ### General response to Reviewer ****fDtg****
>
> We trust that our additional results and expanded explanations address your concerns, further **underscoring the soundness of our work**. We have conducted extensive evaluations on each component of our pipeline, demonstrating the **indispensable nature of each module** we've incorporated. Given the promising results we've achieved on this task, we hope the reviewer would consider increasing the soundness score of our paper

---

### Meta-Review · Area_Chair_hRhp · 2023-09-19

**Recommendation:** 3

**Metareview:**

This paper proposes a new method for zero-shot multimodal QA, MoqaGPT, which extracts answers in each modality and then later fuses answers using LLMs. The proposed method shows strong performance. While the novelty might be limited, overall the proposed method is well-motivated, sound, and strong. I recommend this paper for acceptance.
On the other hand, in the current draft, the main ablations of different models are mostly limited to GPT-4 and ChatGPT, which may introduce issues of data contamination or inability to reproduce the results. I strongly recommend authors include the ablation results with more open-sourced models (for reasoners) as shared in https://openreview.net/forum?id=wrBIS6FOfV&noteId=bPutO2Ja6P in the final version as well. While I understand the scaling of models strongly affects the final performance, due to the nature of such closed and proprietary LLM APIs, it is important to include the results of those open-sourced, reproducible LLM results in the paper.

---

### Decision · Program_Chairs · 2023-10-07

**Decision:**

Accept-Findings

**Comment:**

This paper proposes a new method for zero-shot multimodal QA, MoqaGPT, which extracts answers in each modality and then later fuses answers using LLMs. The proposed method shows strong performance. While the novelty might be limited, overall the proposed method is well-motivated, sound, and strong. I recommend this paper for acceptance.
On the other hand, in the current draft, the main ablations of different models are mostly limited to GPT-4 and ChatGPT, which may introduce issues of data contamination or inability to reproduce the results. I strongly recommend authors include the ablation results with more open-sourced models (for reasoners) as shared in https://openreview.net/forum?id=wrBIS6FOfV&noteId=bPutO2Ja6P in the final version as well. While I understand the scaling of models strongly affects the final performance, due to the nature of such closed and proprietary LLM APIs, it is important to include the results of those open-sourced, reproducible LLM results in the paper.